# Ferroelectric and Relaxor-Ferroelectric Phases Coexisting Boosts Energy Storage Performance in (Bi_0.5_Na_0.5_)TiO_3_-Based Ceramics

**DOI:** 10.3390/molecules29133187

**Published:** 2024-07-04

**Authors:** Yunting Li, Guangrui Lu, Yan Zhao, Rui Zhao, Jiaqi Zhao, Jigong Hao, Wangfeng Bai, Peng Li, Wei Li

**Affiliations:** 1School of Materials Science and Engineering, Liaocheng University, Liaocheng 252059, China; 2310180115@stu.lcu.edu.cn (Y.L.); 2210180108@stu.lcu.edu.cn (G.L.); 2021403623@stu.lcu.edu.cn (Y.Z.); 2021403640@sut.lcu.edu (R.Z.); 2021403612@stu.lcu.edu.cn (J.Z.); haojigong@lcu.edu.cn (J.H.); 2College of Materials and Environmental Engineering, Hangzhou Dianzi University, Hangzhou 310018, China; bwf@hdu.edu.cn

**Keywords:** BNT-based dielectric ceramics, phase coexistence, energy storage properties, temperature and cycling stability

## Abstract

With the intensification of the energy crisis, it is urgent to vigorously develop new environment-friendly energy storage materials. In this work, coexisting ferroelectric and relaxor-ferroelectric phases at a nanoscale were constructed in Sr(Zn_1/3_Nb_2/3_)O_3_ (SZN)-modified (Bi_0.5_Na_0.5_)_0.94_Ba_0.06_TiO_3_ (BNBT) ceramics, simultaneously contributing to large polarization and breakdown electric field and giving rise to a superior energy storage performance. Herein, a high recoverable energy density (W_rec_) of 5.0 J/cm^3^ with a conversion efficiency of 82% at 370 kV/cm, a practical discharged energy density (W_d_) of 1.74 J/cm^3^ at 230 kV/cm, a large power density (P_D_) of 157.84 MW/cm^3^, and an ultrafast discharge speed (t_0.9_) of 40 ns were achieved in the 0.85BNBT-0.15SZN ceramics characterized by the coexistence of a rhombohedral-tetragonal phase (ferroelectric state) and a pseudo-cubic phase (relaxor-ferroelectric state). Furthermore, the 0.85BNBT-0.15SZN ceramics also exhibited excellent temperature stability (25–120 °C) and cycling stability (10^4^ cycles) of their energy storage properties. These results demonstrate the great application potential of 0.85BNBT-0.15SZN ceramics in capacitive pulse energy storage devices.

## 1. Introduction

With the growing demand for advanced electrical power systems, dielectric capacitors, as essential elements for electrostatic energy storage, play a decisive role in high-power applications and pulsed power technologies, owing to their unique advantages of fast charge-and-discharge speed, ultrahigh power density, and excellent stability and reliability [1,2,3]. Compared to polymer-based capacitors [4], the ceramic-based capacitors possess higher capacitance and better temperature stability, and thus, have received increasing attention both in academic research and commercial applications. However, the relatively low recoverable energy density (W_rec_) caused by the low breakdown electric field (E_b_) in dielectric ceramics limits their energy storage applications, for which device miniaturization and system intellectualization are necessary [5,6]. Therefore, a great deal of research has been carried out to explore high-performance dielectric ceramics with high W_rec_ and energy efficiency (η) as well as excellent temperature and cycling stability. In general, the total energy density (W_tot_), the recoverable energy density (W_rec_), and the efficiency (η) of ceramic-based dielectric capacitors can be obtained by the Equations (1)–(3) [7,8]:(1)$$W_{\mathrm{tot}}=\int_{0}^{P_{\max}}E\mathrm{dP}$$
(2)$$W_{\mathrm{rec}}=\int_{P_{r}}^{P_{\max}}E\mathrm{dP}$$
(3)$$\eta =\frac{W_{\mathrm{rec}}}{W_{\mathrm{tot}}}\times 100\%$$
where P_max_ and P_r_ present the maximum polarization and remanent polarization, respectively; E is the applied external electric field. Equations (1)–(3) indicate that a high W_rec_ can be achieved through improving P_max_, reducing P_r_, and simultaneously enhancing E_b_.

Different from linear dielectrics (LDs) and anti-ferroelectrics (AFEs), relaxor ferroelectrics (RFEs) have the combined advantages of a high difference (ΔP = P_max_ − P_r_) and a relatively large breakdown electric field (E_b_), and thus, they possess a pronounced energy storage performance [7,9,10,11,12,13,14]. It is recognized that this pronounced energy storage performance for RFEs is generally strongly related to polar nanoregions (PNRs), which are usually driven by random fields caused by composition disorder disrupting ranged ferroelectric domains [15,16,17,18]. In the nonergodic relaxor (NR) state, the PNRs freeze and the ferroelectric domains with micro-scale generate strong hysteresis, a large P_r_ and inferior energy storage performance. By increasing the content of the composition and/or temperatures, the power and size of the PNRs increases considerably and decreases rapidly, respectively, and quick merging is achieved, resulting in a large reduction in P_r_ [19]. In the dominated ergodic relaxor (ER) state, PNRs with random dipole orientation are easily deflected and ergodic, and the polarization direction and intensity of them can alter with the application of an external electric field [20,21]. Despite the transformation from the RFE to the ferroelectric state with the help of random fields, the long-range ferroelectric domains are disrupted, thus producing disordered PNRs with the removal of electric field [22,23]; that is, P_r_ is significantly suppressed and P_max_ remains relatively high. Accordingly, tailoring RFEs to the dominated ER state using PNRs that generate a small P_r_, a large P_max_, reduced hysteresis, and enhanced thermal stability is the main collaborative optimization strategy (domain engineering), and enhances the energy storage performance, as shown by the slim polarization versus electric field (P (E)) loops with a significantly delayed polarization saturation. Moreover, a higher electric field can drive a larger P_max_, which supports the idea that improving E_b_ by multiple effects is crucial for achieving a high energy density of dielectrics. As a result, simultaneous optimization of the external (sample thickness, electrode shape/size, etc.) and internal (e.g., grain size, defects, and grain orientation) factors has been performed to enhance the E_b_ [24,25]. For instance, a large E_b_ value of 420 kV/cm was obtained in BiFeO_3_-BaTiO_3_-NaNbO_3_ ceramics via reducing the grain size [25]. Hence, tailoring the grain size, which aims to enhance E_b_, was also adopted as a collaborative optimization strategy (i.e. grain size engineering). The decreased grain size means the increased fraction of the grain boundary, which can increase the depletion space charge layers and hence, offers higher potential barriers for charge carriers [26].

It is well known the W_rec_ and η of the typical ferroelectric system (Bi_0.5_Na_0.5_)_0.94_Ba_0.06_TiO_3_ are too small to satisfy the requirements of the system’s practical application, as a result of premature polarization saturation and a large P_r_. In this work, Sr(Zn_1/3_Nb_2/3_)O_3_, selected as an endmember, was introduced to (Bi_0.5_Na_0.5_)_0.94_Ba_0.06_TiO_3_ to partially reduce the ferroelectricity and enhance the relaxation behavior, and eventually, the coexisting ferroelectric and relaxor-ferroelectric phases were constructed, which gave rise to superior energy storage properties (W_rec_ = 5.0 J/cm^3^, η = 82%) in the 0.85BNBT-0.15SZN ceramics.

## 2. Results and Discussion

Figure 1a displays the XRD patterns of (1 − x)BNBT-xSZN (x = 0–0.18) ceramics with different doping concentrations of SZN. All samples exhibit a typical perovskite structure, and not any impurity phase was detected, indicating that the SZN have completely diffused into the BNBT matrix crystal lattice and formed new solid solutions. The enlarged XRD patterns in the 2θ range of 46–47° are depicted in Figure 1b. It is evident that the (200) diffraction peaks gradually move to lower degrees with increasing the SZN content, indicating a lattice expansion, which is due to the substitutions of Sr^2+^ with a relative large ionic radius (R_Sr_^2+^ = 1.44 Å) for Na^+^ (R_Na_^+^ = 1.39 Å) and Bi^3+^ (R_Bi_^3+^ = 1.38 Å) on the A-sites, and the substitutions of Zn^2+^ (R_Zn_^2+^ = 0.74 Å) and Nb^5+^ (R_Nb_^5+^ = 0.64 Å) for Ti^4+^ (R_Ti_^4+^ = 0.605 Å) on the B-sites [27]. The rietveld refinements of the XRD patterns for (1 − x)BNBT-xSZN ceramics, and the TEM selected area electron diffraction (SAED) for representative 0.85BNBT-0.15SZN ceramics are given in Figure 2 to further identify the phase structures. The low values of all the reliability factor of patterns (R_p_ < 7%), the reliability factor of weighted patterns (R_wp_ < 8%), and the goodness-of-fit indicator (χ^2^ < 2%), as shown in Figure 2a–e, indicate that the structural model is valid and the refinement results are consistent with the experimental data. Therefore, we can infer that the crystal structures evolve from R and T phases for the ceramics with x = 0 to coexisting R, T, and C phases for the ceramics with x = 0.08–0.15, and eventually, to the C phase for the ceramics with x = 0.18. The detailed phase volume fractions of the (1 − x)BNBT-xSZN (x = 0–0.18) ceramics are displayed in the insets of Figure 2a–e. Figure 2f and Figure 2g show the SAED patterns along [001]_c_ and [110]_c_ directions, respectively, for the representative sample x = 0.15. The 1/2(ooe) and 1/2(ooo) superlattice spots (where o and e stand for odd and even Miller indices, respectively) were observed in the sample, demonstrating the existence of T and R phases in a pseudo-cubic matrix, which is consistent with the results of the Rietveld refinement of XRD pattern (Figure 2d). It should be pointed out that these superlattice diffractions are not observed in the XRD patterns, as these superlattice diffractions are weak. Combined with the results of XRD refinements and SAED patterns, we can infer that R, T, and pseudo-cubic phases coexist in the x = 0.15 sample. It is worth noting that the desired coexistence of a ferroelectric state (R and T phases) and a relaxor-ferroelectric state (pseudo-cubic phase) constructed by introducing SZN into BNBT ceramics is conducive to enhancing the P_max_ and E_b_ values simultaneously, and thus, results in an excellent energy storage density.

The SEM images of surface morphologies and the corresponding grain size distributions of the (1 − x)BNBT-xSZN ceramics are shown in Figure 3a–e. It can be seen that all the samples present a compact microstructure and clear grain boundary, suggesting good sintering behavior. The average grain size and relative density as a function of SZN doping content are presented in Figure 3f. It is evident that the grain size first increases slightly and then decreases with increasing SZN content. The slight increase in grain size for the ceramics x ≤ 0.10 may be related to the fact that a trace amount of SZN dopant acts as a sintering assistant and raises the kinetic energy of grain growth. However, the consistent reduction of grain size with further increase of SZN doping content (x > 0.10) can be explained as follows. The substitution of larger Sr^2+^ ions for smaller Na^+^ and Bi^3+^ on the A-sites, and the large Zn^2+^ and Nb^5+^ ions for smaller Ti^4+^ ions on the B-sites increases the lattice strain energy and thus, the grain boundary mobility is inhibited [28]. It is worth noting that the relative density of all samples is larger than 97.5%, demonstrating a dense microstructure. It is well known that a dense microstructure and refined grains can substantially increase the breakdown electric field (E_b_) [29,30].

The temperature dependence of permittivity curves is a powerful tool for revealing the composition-induced transition between the ferroelectric and relaxor-ferroelectric states. Figure 4a,b display the temperature dependences of the permittivity (ε) and loss tangent (tanδ) measured at different frequencies for two representative compositions x = 0 and x = 0.15, respectively. It is evident that two dielectric peaks (labeled as T_m_ and T_s_) exist in the ε-T curves. The dielectric peak T_m_ is originated from the transition from the ferroelectric to the paraelectric phase, while the dielectric peak T_s_ is caused by the transformation of ferroelectric rhombohedral to tetragonal PNRs [31]. As compared to the ceramic with x = 0, a strong frequency dependence of T_m_ and an apparent broadening of T_m_ peaks are observed in the ceramic with x = 0.15, demonstrating frequency dispersion and diffused phase transition behaviors, which are the typical characteristics of relaxor ferroelectrics [32]. In order to clearly compare the changes of dielectric properties, the temperature dependence of ε and tanδ curves of the (1 − x)BNBT-xSZN (x = 0–0.18) ceramics at 100 kHz is depicted in Figure 4c. It can be seen that the T_m_ and T_s_ move to a lower temperature and a widened permittivity platform can be found near T_m_ with the increasing of SZN concentration, which may be attributed to the increased disorder degree of A- and B-site ions, the reduced structural stability, and thus, the enhanced relaxor behavior through the introduction of SZN [33,34,35]. Furthermore, the values of room temperature ε, ε_m_ (the maximum of ε at T_m_) and tanδ exhibit a gradual downward tendency with the increase of SZN content, which may be attributed to the enhanced relaxor behavior, and are conducive to enhancing the E_b_. In addition, the relaxor behavior of dielectric ceramics can be quantitatively assessed by the diffusion coefficient (γ), which can be calculated through the modified Curie-Weiss law [36]:(4)$$\frac{1}{\varepsilon }-\frac{1}{\varepsilon_{m}}=\frac{{(T\;-\;T_{m})}^{\gamma }}{C}$$
where ε is the permittivity, ε_m_ is the maximum of ε at T_m_, and C is the Curie constant. The γ value can be obtained from the slope of the linearly fitted curve of ln(1/ε − 1/ε_m_) as a function of ln(T − T_m_), as shown in Figure 4d; the γ increases as the doping content increases and is determined to be 1.96 for the sample with x = 0.15, suggesting a strong relaxor behavior.

Figure 5a displays the bipolar P(E) loops of (1 − x)BNBT-xSZN ceramics measured at 120 kV/cm. The (Bi_0.5_Na_0.5_)_0.94_Ba_0.06_TiO_3_ ceramic (i.e., x = 0) presents a square-like P(E) hysteresis loop with high values of both P_max_ and P_r_, indicating a strong ferroelectric characteristic. With the increase of SZN concentration, both the P_max_ and P_r_ values decrease and the P(E) loops become slim, implying the long-range order in the ferroelectric state is gradually disrupted and transformed into a relaxor-ferroelectric state [37]. Moreover, the E_b_ values are summarized in Figure 5b, and the Weibull distribution plots prove the reliability of the test results of E_b_, as all the shape parameters β (the slope of the fitted lines) are larger than 11 [38,39]. Figure 5c shows the unipolar P(E) loops of the (1 − x)BNBT-xSZN ceramics measured at E_b_; the corresponding energy storage properties (W_rec_ and η) are calculated using Equations (1)–(3) and displayed in Figure 5d. It is worth noting that the optimal energy storage properties (W_rec_ ~ 5.0 J/cm^3^, η ~ 82%) are obtained in the ceramics with x = 0.15 under an electric field of 370 kV/cm. The ferroelectric properties and energy storage performance of (1 − x)BNBT-xSZN ceramics are summarized in Table 1. The comparison of W_rec_ and η between the sample with x = 0.15 obtained in this work and some other representative BNT-based ceramics is presented in Figure 5e [40,41,42,43,44,45,46,47]. One can see that the sample with x = 0.15 exhibits a satisfactory energy storage performance.

The prominent advantages of pulsed power devices are their exceptional power density and fast discharge speed, which exceed those of other conventional power supplies. Therefore, in order to accurately evaluate the practical pulsed charge-and-discharge properties, the under-damped and over-damped discharge current curves of 0.85BNBT-0.15SZN ceramics under various electric fields were measured, and the results are shown in Figure 6. It can be seen from Figure 6a that the peak current (I_max_) increases significantly with the increment of an applied electric field. The current density (C_D_) and power density (P_D_) can be obtained from the under-damped discharge current curves using Equations (5) and (6), respectively:C_D_ = I_max_/S(5)
P_D_ = E I_max_/2S(6)
where S is the electrode area, E is the applied electric field, and I_max_ is the peak current [48]. Figure 6b gives the C_D_ and P_D_ values of the 0.85BNBT-0.15SZN ceramics as a function of electric field. Obviously, the C_D_ and P_D_ values gradually increase with increasing electric field, and the maximum values of 1434.94 A/cm^2^ and 157.84 MW/cm^3^, respectively, are obtained at 220 kV/cm. Figure 6c shows the over-damped discharge current curves (I(t)) of the 0.85BNBT-0.15SZN ceramics under various electric fields. Generally, the discharge energy density (W_d_) can be obtained by integrating the I(t) curves according to Equation (7):(7)$$W_{d}=R\int {I(t)}^{2}\mathrm{dt}/V$$
where R is the load resistance (200 Ω in this work), I is the discharge current, t is the discharge time, and V is the volume of sample. It is obvious that the W_d_ values gradually increase from 0.67 J/cm^3^ to 1.74 J/cm^3^ as the electric field increases from 130 kV/cm to 230 kV/cm. It should be pointed out that the recoverable energy density (W_rec_) obtained by integrating the P(E) loop is usually higher than the discharged energy density (W_d_) measured by the discharge current curve under the same electric field. This phenomenon can be attributed to different testing mechanisms. The P(E) loop test is almost in the millisecond level (i.e., 1–100 Hz), while the charge-and-discharge test is mostly in the order of microseconds or nanoseconds, and the hysteresis effect caused by fast domain switching is more significant [49]. Therefore, the discharged energy density is lower than the recoverable energy storage density. In addition, the discharge speed is also a crucial parameter for the application of pulse power devices. Generally, the discharge speed is determined by the time (t_0.9_) required to release 90% of the total energy density (W_d_) [50]. The discharge time (t_0.9_) is determined to be only 40 ns, proving the ultrafast discharge speed of the ceramic x = 0.15. These results suggest that the 0.85BNBT-0.15SZN ceramics exhibit great potential for applications in high power pulse systems.

To better reveal the microscopic origin of the outstanding energy storage performance of the 0.85BNBT-0.15SZN ceramics, the transmission electron microscopy (TEM) measurements were carried out. Figure 7a,b show the bright field TEM (BF-TEM) images of two representative grains in the 0.85BNBT-0.15SZN ceramics. Of particular interest is that both lamellar-shaped domains, which are the signature of the ferroelectric phase [51], and blotchy domains, which correspond to the PNRs, are observed in the ceramics with x = 0.15. Figure 7c,d shows the HR-TEM lattice fringe images obtained from the areas of the lamellar-shaped domains (Figure 7a) and the polar nanoregions (Figure 7b), respectively, indicating the ordered arrangement of atoms and the fine crystalline quality. To further elucidate the local polarization fluctuation behavior of the ferroelectric domains and PNRs, the inverse FFT patterns of the selected polar regions (30 × 30 nm^2^, marked by the red box in Figure 7c,d) are given in Figure 7e and Figure 7f, respectively. The yellow regions in the inverse FFT images represent the local polar regions [52,53]. Interestingly, both areas display a remarkable local structural heterogeneity characteristic, which is caused by the random field generating due to the cations with different valences and sizes occupying the A- (Na^+^, Bi^3+^, Ba^2+^, and Sr^2+^) and B- (Ti^4+^, Zn^2+^, and Nb^5+^) sites in the unit cell [54]. It is well known the local heterogeneity can impede the coherent length of dipoles and disrupt the long-range polar correlation [20,55]. Therefore, one can see the inverse FFT images (Figure 7e) obtained from the lamellar-shaped domains display larger sized and more tightly connected polar regions as compared with those (Figure 7f) obtained from polar nano-regions, providing solid evidence that SZN doping disrupted the ferroelectric long-range order, reduced the size of local polar regions, and formed PNRs. It is worth noting that the lamellar-shaped domains (i.e., the ferroelectric phase) can help to enhance the P_max_ under an external electric field, while the PNRs (i.e., the relaxor-ferroelectric phase) can delay polarization saturation to realize a high E_b_ in the dielectric ceramics. Hence, we can infer that the coexistence of ferroelectric and relaxor-ferroelectric phases gives rise to a high energy storage density (W_rec_). Furthermore, the not-too-high efficiency (η < 90%) may be closely associated with the switching of the residual lamellar-shaped ferroelectric domains under an external electric field, which usually results in a hysteretic P(E) loop.

The temperature and cycle stability are crucial for energy storage applications. The temperature and cycle dependent unipolar P(E) loops are given in Figure 8a,b, and corresponding W_rec_ and η values of the x = 0.15 ceramics under an electric field of 200 kV/cm are given in Figure 8c and Figure 8d, respectively. Apparently, all the P(E) loops do not change significantly and the P_r_ and P_max_ values present minimal variation, leading to slight fluctuations of W_rec_ (<13%) and η (<11%) in the temperature range of 25–120 °C and incredibly small variations of W_rec_ (<2%) and η (<4%) during 10^4^ cycles for the ceramics with x = 0.15, suggesting superior temperature stability and fatigue-resistant behavior of energy storage performance. Figure 9a and 9b show the over-damped discharge current curves and W_d_(t) curves at different temperatures under 150 kV/cm for the x = 0.15 ceramics, respectively. Meanwhile, the corresponding W_d_ and t_0.9_ values under various temperatures are calculated and summarized in Figure 9c and 9d, respectively. It should be noted here that the W_d_ and η decrease slightly, from 1.0 J/cm^3^ to 0.9 J/cm^3^, and from 52 ns to 41 ns when the temperature rises from 25 °C to 120 °C, exhibiting variations of less than 10% and 21%, respectively, which means the 0.85BNBT-0.15SZN ceramics have an excellent temperature stability of charge-and-discharge properties. The outstanding temperature stability may be closely related to the gentle fluctuation of permittivity (ε) with respect to temperature, as shown in Figure 4c. The reliable temperature stability of the energy storage performance widens the temperature usage range of the 0.85BNBT-0.15SZN ceramics in dielectric capacitors.

## 3. Materials and Methods

### 3.1. Materials and Synthesis

(1 − x)(Bi_0.5_Na_0.5_)_0.94_Ba_0.06_TiO_3_-xSr(Zn_1/3_Nb_2/3_)O_3_ (abbreviated as (1 − x)BNBT-xSZN, x = 0, 0.08, 0.10, 0.15 and 0.18) ceramics were prepared via the conventional solid-state reaction method. High-purity raw materials of Bi_2_O_3_ (99%), Na_2_CO_3_ (99.8%), TiO_2_ (99.9%), BaCO_3_ (99.99%), SrCO_3_ (99.95%), ZnO (99.99%), and Nb_2_O_5_ (99.9%), weighted according to the stoichiometric chemical formula, were mixed by ball milling in anhydrous ethanol for 24 h. After drying the slurry, the mixed powders were calcined at 850 °C for 5 h, and then milled again for 24 h. Subsequently, the dried powders were mixed with polyvinyl alcohol binder and then were pressed into disks with a diameter of 10 mm. Finally, the green bodies were sintered at 1060–1120 °C for 3 h to form densified ceramics.

### 3.2. Characterization

The phase structure of (1 − x)BNBT-xSZN ceramics was identified by X-ray diffractometer (XRD, TD-3700, Dandong Tongda Technology Co. Ltd., Dandong, China). Structural refinement was performed using the Rietveld refinement program GSAS. The microstructure was characterized by means of field-emission scanning electron microscope (FE-SEM, Carl Zeiss, Oberkochen, Germany). The grain size distributions of the samples were obtained using Nano Measurer software (version number 1.2). The domain morphology was observed by using transmission electron microscope (TEM, FEI Talos F200X, Waltham, MA, USA) operated at 200 kV. The temperature dependence of dielectric properties was tested by a precision LCR meter (Agilent E4980A, Santa Clara, CA, USA). For energy storage measurements, the samples were polished to a thickness of ~0.06 mm, and then both sides were sputtered gold electrodes with a diameter of 0.5 mm. The polarization-electric field (P-E) hysteresis loops were measured by a ferroelectric analyzer (RT1-Premier II, Radiant Technologies Inc., Alpharetta, GA, USA) at room temperature in a frequency of 10 Hz. A commercial charge-discharge system (CFD-003, Tongguo Technology, Shanghai, China) was employed to measure the practical energy release performance.

## 4. Conclusions

In summary, (1 − x)BNBT-xSZN (x = 0, 0.08, 0.10, 0.15 and 0.18) dielectric ceramics were synthesized by a conventional solid-state reaction method. The optimal energy storage properties, with a high W_rec_ of 5.0 J/cm^3^ and an acceptable η of 82% under 370 kV/cm, were obtained for the x = 0.15 ceramics. The XRD refinement and TEM measurements demonstrate that the introduction of SZN disrupted the long-range ferroelectric order, driving the formation of PNRs and eventually resulting in the coexistence of ferroelectric and relaxor-ferroelectric phases in the x = 0.15 ceramics. The coexistence of ferroelectric and relaxor-ferroelectric phases contributes to simultaneously enhanced P_max_ and E_b_ values, and thus, gives rise to excellent energy storage properties. Meanwhile, the 0.85BNBT-0.15SZN ceramics exhibited a large power density (P_D_ = 157.84 MW/cm^3^ under 220 kV/cm) and ultrafast discharge speed (t_0.9_ = 40 ns). Moreover, an outstanding temperature stability of energy storage and charge-and-discharge properties, as well as a cycle stability, were also achieved in the 0.85BNBT-0.15SZN ceramics, which is of great importance for the operation of energy storage dielectric materials in harsh environments. All the merits demonstrate that the 0.85BNBT-0.15SZN ceramic is a promising candidate for high-power energy storage devices.

## Figures and Tables

**Figure 1 molecules-29-03187-f001:**
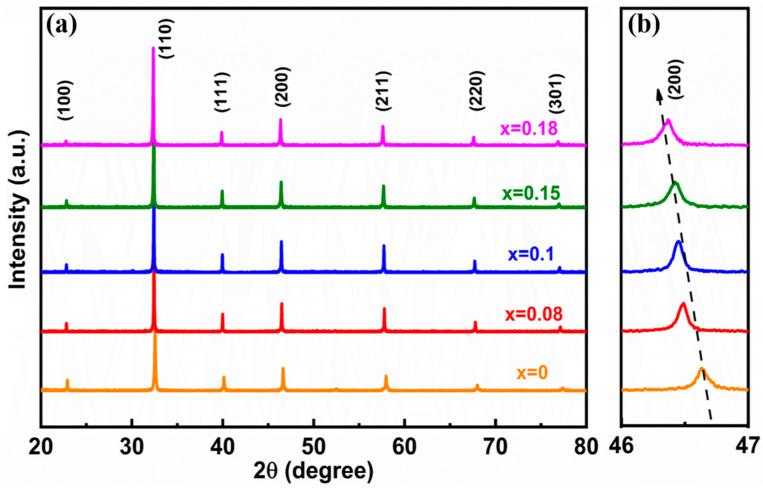
XRD patterns of (1 − x)BNBT-xSZN ceramics in the 2θ range of (**a**) 20–80° and (**b**) 46–47°.

**Figure 2 molecules-29-03187-f002:**
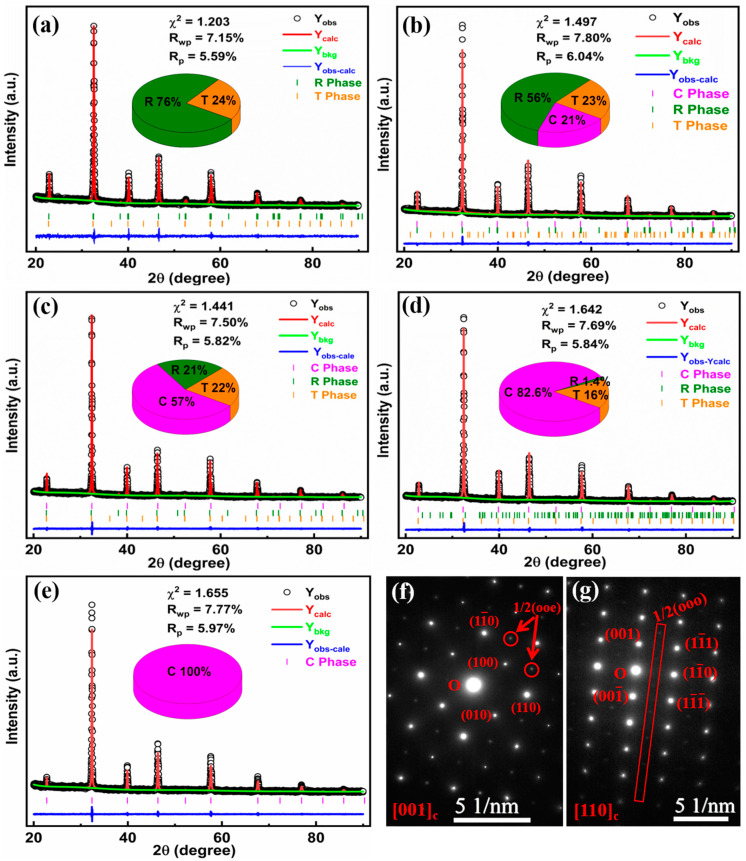
Rietveld refinement of XRD patterns of (1 − x)BNBT-xSZN ceramics with (**a**) x = 0, (**b**) x = 0.08, (**c**) x = 0.10, (**d**) x = 0.15, and (**e**) x = 0.18; SAED patterns along [001]_c_ (**f**) and [110]_c_ (**g**) directions for the sample with x = 0.15.

**Figure 3 molecules-29-03187-f003:**
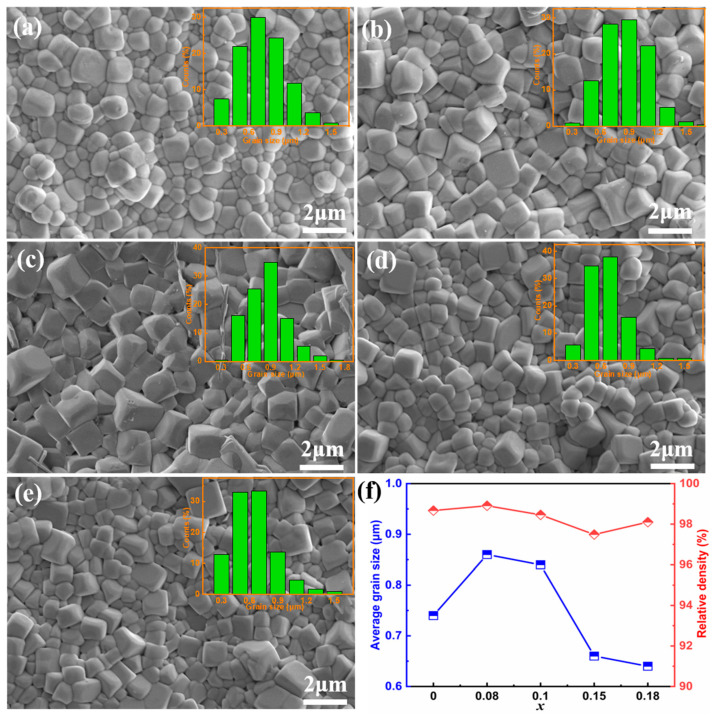
SEM images of (1 − x)BNBT-xSZN ceramics: (**a**) x = 0, (**b**) x = 0.08, (**c**) x = 0.10, (**d**) x = 0.15, and (**e**) x = 0.18. The insets in Figure 3a–e show the grain size distributions. (**f**) The average grain size and relative density as a function of SZN doping content.

**Figure 4 molecules-29-03187-f004:**
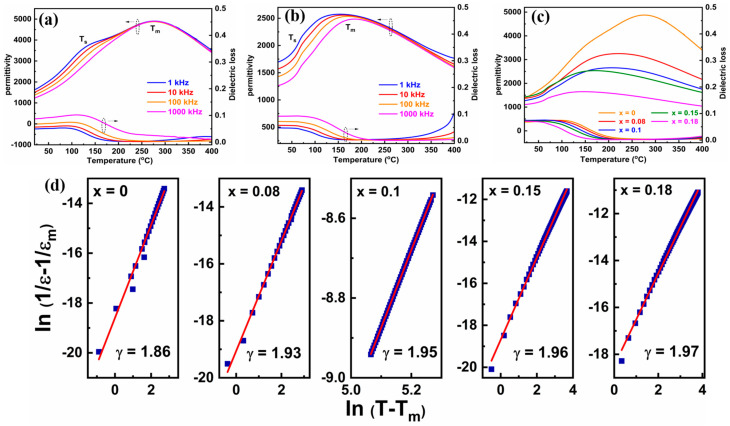
Temperature-dependent permittivity (ε-T) and dielectric loss (tanδ-T) at different frequencies for the (1 − x)BNBT-xSZN ceramics (**a**) x = 0 and (**b**) x = 0.15. (**c**) Temperature dependent ε and tanδ for the (1 − x)BNBT-xSZN ceramics at 100 kHz. (**d**) ln(1/ε − 1/ε_m_) versus ln(T − T_m_) for the (1 − x)BNBT-xSZN ceramics.

**Figure 5 molecules-29-03187-f005:**
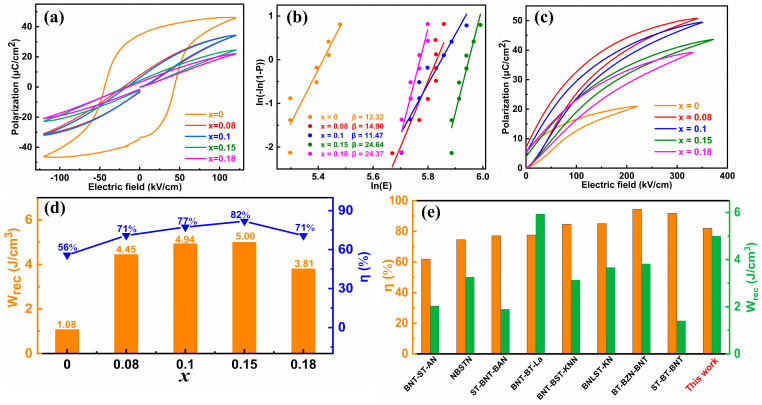
(**a**) Bipolar P(E) hysteresis loops measured at 120 kV/cm and 10 Hz. (**b**) Weibull distribution of breakdown electric fields. (**c**) Unipolar P(E) hysteresis loops measured at E_b_ and 10 Hz. (**d**) W_rec_ and η of (1 − x)BNBT-xSZN (x = 0–0.18) ceramics calculated at their breakdown electric fields. (**e**) Comparison of W_rec_ and η between the 0.85BNBT-0.15SZN sample and some other representative BNT-based lead-free ceramics.

**Figure 6 molecules-29-03187-f006:**
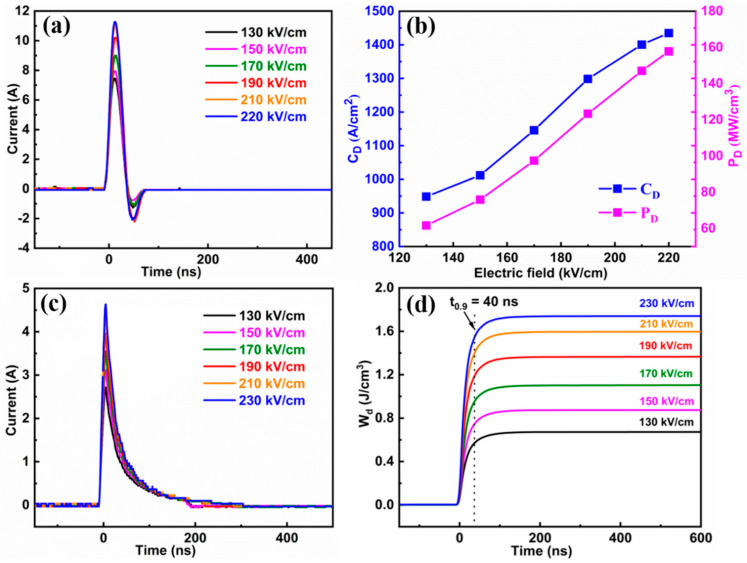
(**a**) Under-damped pulse discharge current curves of 0.85BNBT-0.15SZN ceramics at various electric fields. (**b**) Variation of C_D_ and P_D_ as a function of the applied electric field. (**c**) Over-damped pulse discharge current curves of 0.85BNBT-0.15SZN ceramics at various electric fields. (**d**) Discharge energy density (W_d_) as a function of time for 0.85BNBT-0.15SZN ceramics.

**Figure 7 molecules-29-03187-f007:**
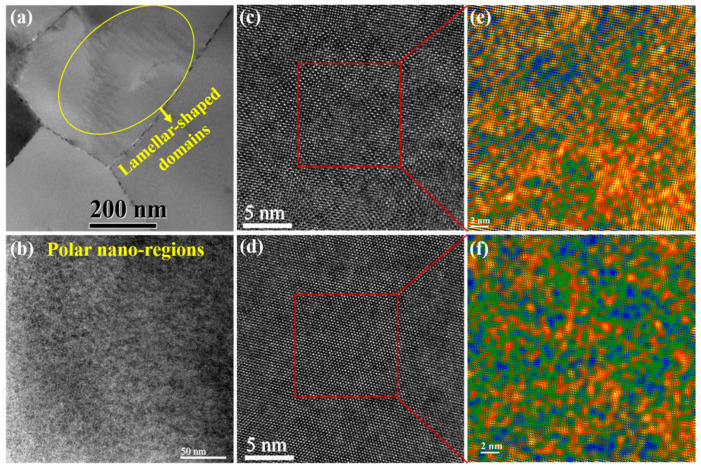
(**a**,**b**) TEM bright-field images of two representative grains for the sample with x = 0.15. (**c**,**d**) HR-TEM lattice fringe images. (**e**,**f**) Inverse fast Fourier transform (IFFT) images converted from the areas marked with the red square lines in Figure 7c and Figure 7d, respectively, for 0.85BNBT-0.15SZN ceramics.

**Figure 8 molecules-29-03187-f008:**
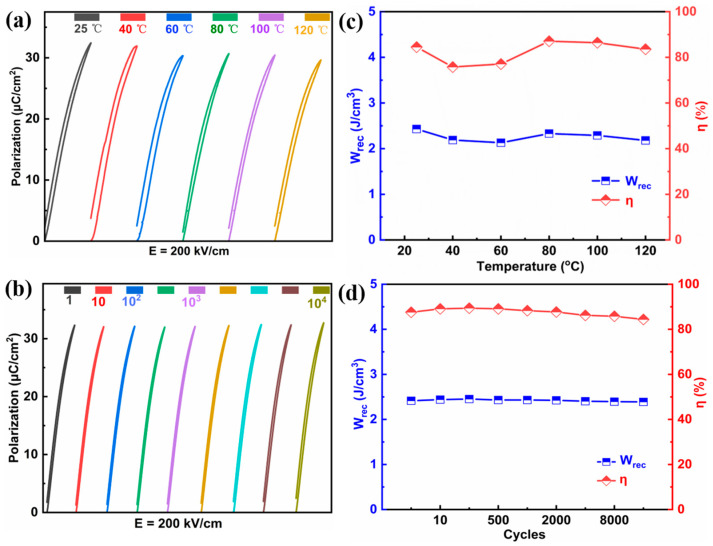
(**a**,**b**) Unipolar P-E loops and (**c**,**d**) corresponding W_rec_ and η values as functions of temperature and cycles for the ceramics with x = 0.15 under an electric field of 200 kV/cm.

**Figure 9 molecules-29-03187-f009:**
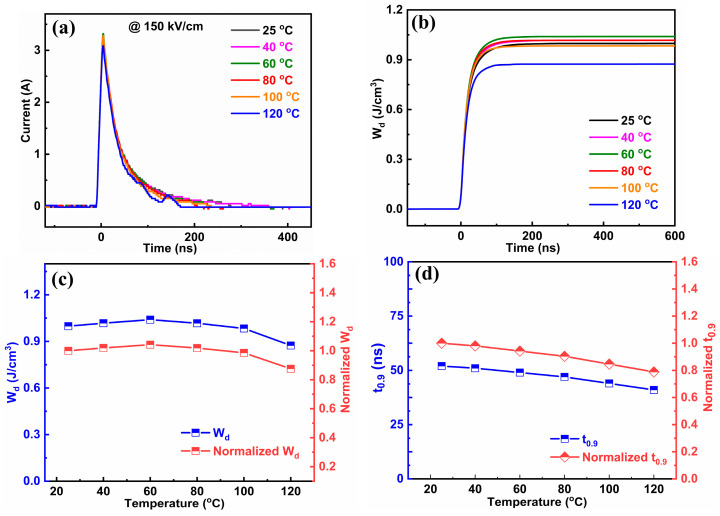
(**a**) Over-damped discharge current curves, (**b**) W_d_ dependence of time measured at different temperatures, and variations of (**c**) W_d_ and (**d**) t_0.9_ as a function of temperature for the ceramics with x = 0.15 under an electric field of 150 kV/cm.

**Table 1 molecules-29-03187-t001:** Summary of ferroelectric properties and energy storage performance of (1 − x)BNBT-xSZN ceramics.

*x*	P_r_ (μC/cm^2^)	P_max_ (μC/cm^2^)	E_c_ (kV/cm)	E_b_ (kV/cm)	W_rec_ (J/cm^3^)	η
0	5.37	20.92	43.30	226	1.08	56%
0.08	7.25	50.77	20.57	342	4.45	71%
0.10	5.21	49.46	16.22	347	4.94	77%
0.15	3.96	43.42	17.75	383	5.00	82%
0.18	5.44	39.15	14.43	321	3.81	71%

## Data Availability

Data are contained within the article.

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
