# Peer review of "Ferroelectric and Relaxor-Ferroelectric Phases Coexisting Boosts Energy Storage Performance in (Bi0.5Na0.5)TiO3-Based Ceramics"

_molecules, 2024, doi:10.3390/molecules29133187_

Round 1

Reviewer 1 Report

Comments and Suggestions for Authors

The present paper sound novel enough, however, the level of the previous plagiarism (46%) is very high. The paper can be accepted after the text is re-written to avoid it.  

Author Response

Comments 1: The present paper sound novel enough, however, the level of the previous plagiarism (46%) is very high. The paper can be accepted after the text is re-written to avoid it.

Response 1: Thank you very much for your kind work and positive comments. We have re-written the paper to reduce the duplication rate with previous work. However, it should be noted here that the repetition of some specialized words is inevitable.

We have studied comments carefully and tried our best to improve the manuscript. We appreciate for reviewer’s warm work earnestly, and hope that the correction will meet with approval.

Once again, thank you very much for your kind work and constructive comments.

Reviewer 2 Report

Comments and Suggestions for Authors

The manuscript by Li et al. entitled "Ferroelectric and relaxor-ferroelectric phases coexisting boosts energy storage performance in (Bi0.5Na0.5)TiO3-based ceramics," presented structural, microstructural, and electrical properties. Paper looks to me very well organized, and the concepts have been clearly explained. However, there are some minor corrections should be addressed to improve the quality of the paper before accepting in Molecules.

1.      In the abstract, the authors stated that “the coexisting ferroelectric and relaxor-ferroelectric phases at a nano-scale were constructed in Sr(Zn1/3Nb2/3)O3 (SZN) modified (Bi0.5Na0.5)0.94Ba0.06TiO3 (BNBT) ceramics, which contribute simultaneously to large polarization and breakdown electric field, and give rise to superior energy storage performance.” Are enhanced energy storage properties due to the formation of highly-dynamic PNRs (FE to RFE transformation) or coexisting both ferroelectric and relaxor-ferroelectric phases? If yes, is there any evidence for coexisting ferroelectric and relaxor-ferroelectric phases?

2.      Figure 4d is discussed with no reasons for the relaxor behavior in the samples; the author should add more effort to justify the relaxor behavior of the results. I suggest to present a modified Curie-Weiss law analysis for all the samples, if not at least for reference and the best sample (presented only for x=0.15).

3.      It is possible to calculate the relative density values of each sample to prove that all samples are dense microstructure.

4.      I suggest the author list all values of ferroelectric properties (Pr, Pmax, Ec and Eb) and energy storage performance (energy density and efficiency) for all compositions in a Table; it is easier to compare all properties with increasing x composition.

5.      The author should discuss the cycle stability is an important property necessary for energy storage applications.

6.      There is no comparison of investigated properties with recent or earlier reports.

Author Response

Thank you very much for taking the time to review this manuscript. We appreciate reviewer very much for their positive and constructive comments and suggestions on our manuscript entitled “Ferroelectric and relaxor-ferroelectric phases coexisting boosts energy storage performance in (Bi0.5Na0.5)TiO3-based ceramics” (Manuscript ID: molecules-3067397). We have studied reviewer’s comments carefully and have tried our best to revise our manuscript according to the comments. Please find the detailed responses below and the corresponding revisions/corrections highlighted/in track changes in the re-submitted files.

Comments 1: In the abstract, the authors stated that “the coexisting ferroelectric and relaxor-ferroelectric phases at a nano-scale were constructed in Sr(Zn1/3Nb2/3)O3 (SZN) modified (Bi0.5Na0.5)0.94Ba0.06TiO3 (BNBT) ceramics, which contribute simultaneously to large polarization and breakdown electric field, and give rise to superior energy storage performance.” Are enhanced energy storage properties due to the formation of highly-dynamic PNRs (FE to RFE transformation) or coexisting both ferroelectric and relaxor-ferroelectric phases? If yes, is there any evidence for coexisting ferroelectric and relaxor-ferroelectric phases?

Response 1: Thank you very much for your constructive comment and suggestion. The enhanced energy storage properties are due to the coexisting of both ferroelectric and relaxor-ferroelectric phases, which is confirmed by TEM bright-field images (Fig. 1a, b). It can be clearly seen that both the lamellar-shaped domains and blotchy domains are coexisted in the sample with x = 0.15. Generally, the lamellar-shaped domains are the signature of ferroelectric phase, and the blotchy domains are the hallmark of relaxor ferroelectric phase. The ferroelectric phase is conducive to increase the polarization, and the relaxor ferroelectric phase helps to boost the breakdown electric field, thus giving rise to superior energy storage properties of the sample with x = 0.15.

Fig. 1 TEM bright-field images of two representative grains for the sample with x = 0.15.

Comments 2: Figure 4d is discussed with no reasons for the relaxor behavior in the samples; the author should add more effort to justify the relaxor behavior of the results. I suggest to present a modified Curie-Weiss law analysis for all the samples, if not at least for reference and the best sample (presented only for x = 0.15).

Response 2: Thank you very much for your constructive suggestion. It is really true as reviewer suggested that the dielectric relaxor behavior can be analyzed by the modified Curie-Weiss law:

where εm is the maximum permittivity, Tm is the temperature corresponding to the maximum permittivity, C is the Curie constant and γ is the diffuseness degree. The γ = 1 represents typical ferroelectric and γ = 2 means a relaxor characteristic. The plots of ln(1/ε-1/εm) versus ln(T-Tm) for all the samples based on the modified Curie-Weiss law are shown in Fig. 2, which have also been added to Figure 4d in the revised manuscript. (Page 6, paragraph 1, and line 165-171)

Fig. 2 ln(1/ε-1/εm) versus ln(T-Tm) for (1-x)BNBT-xSZN ceramics.

Comments 3: It is possible to calculate the relative density values of each sample to prove that all samples are dense microstructure.

Response 3: Thank you very much for your constructive suggestion. The relative density of (1-x)BNBT-xSZN ceramics were tested utilizing the Archimedes drainage method. The relative density of each sample is shown in Fig. 3 and has been added to Fig. 3f in the revised manuscript. It is worth noting that the relative density of all samples is larger than 97.5%, demonstrating a dense microstructure. (Page 5, line 143)

Fig. 3 Average grain size and relative density as a function of SZN doping content x.

Comments 4: I suggest the author list all values of ferroelectric properties (Pr, Pmax, Ec and Eb) and energy storage performance (energy density and efficiency) for all compositions in a Table; it is easier to compare all properties with increasing x composition.

Response 4: Thank you very much for your constructive comment. It is really true as reviewer suggested that it is easier to compare all properties with increasing x composition in a Table. The Table has been added to the revised manuscript. (Page 7, line 200-201)

Table 1 Summary of ferroelectric properties and energy storage performance for (1−x)BNBT-xSZN ceramics

x Pr (μC/cm2) Pmax (μC/cm2) Ec (kV/cm) Eb (kV/cm) Wrec (J/cm3) η
0 5.37 20.92 43.30 226 1.08 55.76%
0.08 7.25 50.77 20.57 342 4.45 70.87%
0.10 5.21 49.46 16.22 347 4.94 77.29%
0.15 3.96 43.42 17.75 383 5.00 81.80%
0.18 5.44 39.15 14.43 321 3.81 70.72%

Comments 5: The author should discuss the cycle stability is an important property necessary for energy storage applications.

Response 5: Thank you very much for your constructive comment. This comment is valuable and very helpful for improving our paper. It is really true as reviewer suggested that the cycle stability is an important property for energy storage applications. The unipolar P-E loops and corresponding Wrec and η values as a function of cycle numbers for the sample of x = 0.15 were performed and shown in Fig. 4. Apparently, all the P (E) loops do not change significantly and the Pr and Pmax values present minimal variation, leading to slight fluctuations of Wrec (< 2%) and η (< 4%) during 104 cycles for the ceramic with x = 0.15, suggesting a fatigue-resistant behavior of energy storage performance. The cycle stability has been added to the revised manuscript. (Page 11, line 297-298)

Fig. 4 (a) Unipolar P-E loops and (b) corresponding Wrec and h values as a function of cycle numbers for the sample with x = 0.15 under an electric field of 200 kV/cm.

Comments 6: There is no comparison of investigated properties with recent or earlier reports.

Response 6: Thank you very much for your constructive comment. According to the reviewer’s suggestion, we have compared the energy storage properties (Wrec, η) of the sample with x = 0.15 obtained in this work with those BNT-based energy storage ceramics reported in the literature [1-8], as shown in Fig. 5. As compared to those BNT-based energy storage ceramics reported in the literature, one can see that the sample with x = 0.15 exhibit a satisfactory energy storage performance in Wrec and η values. A comparison of energy storage properties between the samples with x = 0.15 and those reported in the literature has been added to the revised manuscript. (Page 7, line 198-199)

Fig. 5 Comparison of the energy storage properties between the 0.85BNBT-0.15SZN sample and some other representative BNT-based lead-free ceramics.

References:

[1] W. G. Ma, Y. W. Zhu, M. A. Marwat, P. Y. Fan, B. Xie, D. Salamon, Z. G. Ye, H. B. Zhang, Enhanced energy-storage performance with excellent stability under low electric fields in BNT-ST relaxor ferroelectric ceramics, J. Mater. Chem. C, 2019, 7, 281-288.

[2] B. K. Chu, J. G. Hao, P. Li, Y. C. Li, W. Li, L. M. Zheng, H. R. Zeng, High-energy storage properties over a broad temperature range in La-modified BNT-based lead-free ceramics, ACS Appl. Mater. Interfaces, 2022, 14, 19683-19696.

[3] X. C. Wang, Y. Lu, P. Li, J. Du, P. Fu, J. G. Hao, W. Li, Achieving High Energy Storage Performance under a Low Electric Field in KNbO3-Doped BNT-Based Ceramics, Inorg. Chem., 2024, 63, 7080-7088.

[4] Z. B. Pan, D. Hu, Y. Zhang, J. J. Liu, B. Shen, J. W. Zhai, Achieving high discharge energy density and efficiency with NBT-based ceramics for application in capacitors, J. Mater. Chem. C, 2019, 7, 4072-4078.

[5] Y. S. Zhang, W. H. Li, X. G. Tang, K. Meng, S. Y. Zhang, X. Z. Xiao, X. B. Guo, Y. P. Jiang, Z. H. Tang, Energy storage and charge-discharge performance of B-site doped NBT-based lead-free ceramics, J. Alloy. Compd., 2022, 911, 165074.

[6] H. B. Yang, F. Yan, Y. Lin, T. Wang, L. He, F. Wang, A lead free relaxation and high energy storage efficiency ceramics for energy storage applications, J. Alloy. Compd., 2017, 710, 436-445.

[7] F. Yan, H. B. Yang, Y. Lin, T. Wang, Dielectric and ferroelectric properties of SrTiO3-Bi0.5Na0.5TiO3-BaAl0.5Nb0.5O3 lead-free ceramics for high-energy-storage applications, Inorg. Chem., 2017, 56, 13510-13516.

[8] Y. X. Yan, W. J. Qin, X. Y. Wang, Z. M. Li, D. Y. Zhang, M. L. Zhang, Y. H. Xu, L. Jin, Enhanced energy storage in temperature stable Bi0.5Na0.5TiO3-modified BaTiO3-Bi(Zn2/3Nb1/3)O3 ceramics, Ceram. Int., 2022, 48, 36478-36489.

Once again, thank you very much for your kind work and constructive comments.

Reviewer 3 Report

Comments and Suggestions for Authors

1.      It would be valuable to explain how grain size distributions were obtained.

2.      line 138: You can add ref. to "It is well known that dense microstructure and refined grains can substantially increase the breakdown electric field (Eb)".

3.      line 145 and Fig.4: I suggest replacing the "dielectric constant" with the permittivity (symbol εr).

4.      line 148: Explain the symbol "εr - T" and "dielectric peak".

5.      line 148: Explain the "dielectric constant platform".

6.      line 167: There is "εm is the maximum of εr"; you should add "at Tm".

7.      In the Fig. 4 captions, explain (εr - T) and (tanδ - T).

8.      The graph labels in Fig. 4 and Fig. 9 are too close to each other.

9.      line 177 and others: I suggest replacing P – Ewith P (E). The same for , (I - t) – line 208.

10.  Add ref. to Weibull distribution of breakdown electric fields.

11.  How do you explain the P jump for E =0 in Fig. 5a?

Author Response

Thank you very much for taking the time to review this manuscript. We appreciate reviewer very much for their positive and constructive comments and suggestions on our manuscript entitled “Ferroelectric and relaxor-ferroelectric phases coexisting boosts energy storage performance in (Bi0.5Na0.5)TiO3-based ceramics” (Manuscript ID: molecules-3067397). We have studied reviewer’s comments carefully and have tried our best to revise our manuscript according to the comments. Please find the detailed responses below and the corresponding revisions/corrections highlighted/in track changes in the re-submitted files.

Comments 1: It would be valuable to explain how grain size distributions were obtained.

Response 1: Thank you very much for your constructive comment and suggestion. In this work, the grain size distributions of the samples were obtained using Nano Measurer software, which is a professional tool for measuring grain size distribution. These comments have been added to the section of Materials and Methods. (Page 12, line 319-320)

Comments 2: line 138: You can add ref. to “It is well known that dense microstructure and refined grains can substantially increase the breakdown electric field (Eb)”.

Response 2: Thank you very much for your constructive suggestion, which is helpful for improving our paper. According to the reviewer’s suggestion, I have added several related references to “It is well known that dense microstructure and refined grains can substantially increase the breakdown electric field (Eb)”. (Page 5,line 139)

References:

[1] L. T. Yang, X. Kong, F. Li, H. Hao, Z. X. Cheng, H. X. Liu, J. F. Li, S. J. Zhang, Perovskite lead-free dielectrics for energy storage applications, Prog. Mater. Sci., 2019, 102, 72-108.

[2] R. R Kang, Z. P. Wang, W. J. Yang, X. P. Zhu, L. Q. He, Y. F. Gao, J. T. Zhao, P. Shi, Y. Y. Zhao, P. Mao, Enhanced energy storage performance in Sr0.7La0.2Zr0.15Ti0.85O3-modified Bi0.5Na0.5TiO3 ceramics via constructing local phase coexistence, Chem. Eng. J., 2022, 446, 137105.

Comments 3: line 145 and Fig.4: I suggest replacing the “dielectric constant” with the permittivity (symbol εr).

Response 3: Thank you very much for your constructive suggestion. We have made correction according to the reviewer’s suggestion in the revised manuscript. (Page 5-6)

Comments 4: line 148: Explain the symbol “εr-T” and “dielectric peak”.

Response 4: Thank you very much for your constructive comment. The symbol “εr-T” means permittivity as a function of temperature. “dielectric peak” refers to the maximum of permittivity within the tested temperature range. 

Comments 5: line 157: Explain the “dielectric constant platform”.

Response 5: Thank you very much for your constructive comment. The “dielectric constant platform” means a gradual variation of permittivity with respect to temperature or a maximum of permittivity over a broad temperature range, which is a typical characteristic of relaxor ferroelectrics.

Comments 6: line 167: There is “εm is the maximum of εr”; you should add “at Tm”.

Response 6: Thank you very much for your constructive suggestion. According to the reviewer’s suggestion, we have corrected the relevant description in the revised manuscript. (Page 5-6)

Comments 7: In the Fig. 4 captions, explain (εr-T) and (tanδ-T).

Response 7: Thank you very much for your valuable comment. εr-T means permittivity as a function of temperature, and tanδ-T means dielectric loss as a function of temperature. In other words, εr-T and tanδ-T refer to the variations of permittivity and dielectric loss with respect to temperature, respectively.

Comments 8: The graph labels in Fig. 4 and Fig. 9 are too close to each other.

Response 8: Thank you very much for your comment. The reviewer’s comment is helpful for improving our paper. We have made correction in the revised manuscript. (Page 6 and 11)

Comments 9: line 177 and others: I suggest replacing P – E with P (E). The same for, (I - t) – line 208.

Response 9: Thank you very much for your constructive and helpful suggestion. The P-E and I-t are replaced by P(E) and I(t), respectively, in the revised manuscript. (Page 6-8)

Comments 10: Add ref. to Weibull distribution of breakdown electric fields.

Response 10: Thank you very much for your constructive comment and suggestion. We have added references to Weibull distribution of breakdown electric field in the revised manuscript. (Page 15, line 458-461)

References:

[1] L. A. Dissado, J. C. Fothergill, S. V. Wolfe, R. M. Hill, Weibull Statistics in Dielectric Breakdown; Theoretical Basis, Applications and Implications, IEEE T. El. In., 1984, 3, 227-233.

[2] D. Fabiani, L. Simoni, Discussion on application of the Weibull distribution to electrical breakdown of insulating materials, IEEE T. Dielect. El. In., 2005, 12, 11-16.

Comments 11: How do you explain the P jump for E=0 in Fig. 5a?

Response 11: Thank you very much for your constructive comment. This is an interesting question. The sample with x=0 exhibits strong ferroelectric characteristic with large remnant polarization (Pr) and saturated polarization (Pmax), while with the increase of SZN doping concentration, the ferroelectric properties decrease sharply and the relaxation characteristic boosts significantly. It is well known that the dipoles arranged along the electric field direction will return to a disordered state after removing the external electric field, and thus the values of polarization (P) exhibit a jump at E = 0. It should be noted here that this phenomenon is commonly observed in chemical composition modified BNT-based dielectric ceramics.

We have studied comments carefully and tried our best to improve the manuscript. We appreciate for reviewer’s warm work earnestly, and hope that the correction will meet with approval.

Once again, thank you very much for your kind work and constructive comments.